# Pleiotropy of PP2A Phosphatases in Cancer with a Focus on Glioblastoma *IDH* Wildtype

**DOI:** 10.3390/cancers14215227

**Published:** 2022-10-25

**Authors:** Elham Kashani, Erik Vassella

**Affiliations:** 1Institute of Pathology, University of Bern, 3008 Bern, Switzerland; 2Graduate School for Cellular and Biomedical Sciences, University of Bern, 3012 Bern, Switzerland

**Keywords:** Ser/Thr phosphatase, signaling pathway, brain tumors, glioblastoma, therapy

## Abstract

**Simple Summary:**

GBM is a highly heterogeneous tumor owing to incredible cellular plasticity and adaptation to hypoxia, nutrient availability and therapy. This plasticity is in part due to the dynamic switch of oncogenic pathways that drive GBM progression, generating subclones with distinct alterations in pro-survival signaling pathways. Consequently, despite addiction to specific oncogenic mutations in some GBM subclones, targeted therapies with small molecule kinase inhibitors have very limited efficacy due to the rapid outgrowth of other subclones with a different set of driver mutations. PP2A can simultaneously target many integral inducers of these driver pathways including EGFR, NF-κB, and Wnt kinases. Therefore, PP2A modulation has potential as an adjuvant therapy, as it can drive plastic GBM cells into a dead-end in which they no longer can bear the effects of treatment due to lack of alternative escape pathways.

**Abstract:**

Serine/Threonine protein phosphatase 2A (PP2A) is a heterotrimeric (or occasionally, heterodimeric) phosphatase with pleiotropic functions and ubiquitous expression. Despite the fact that they all contribute to protein dephosphorylation, multiple PP2A complexes exist which differ considerably by their subcellular localization and their substrate specificity, suggesting diverse PP2A functions. PP2A complex formation is tightly regulated by means of gene expression regulation by transcription factors, microRNAs, and post-translational modifications. Furthermore, a constant competition between PP2A regulatory subunits is taking place dynamically and depending on the spatiotemporal circumstance; many of the integral subunits can outcompete the rest, subjecting them to proteolysis. PP2A modulation is especially important in the context of brain tumors due to its ability to modulate distinct glioma-promoting signal transduction pathways, such as PI3K/Akt, Wnt, Ras, NF-κb, etc. Furthermore, PP2A is also implicated in DNA repair and survival pathways that are activated upon treatment of glioma cells with chemo-radiation. Depending on the cancer cell type, preclinical studies have shown some promise in utilising PP2A activator or PP2A inhibitors to overcome therapy resistance. This review has a special focus on “glioblastoma, IDH wild-type” (GBM) tumors, for which the therapy options have limited efficacy, and tumor relapse is inevitable.

## 1. PP2A Forms Multiple Complexes with Distinct Functions

Protein phosphorylation, the most common posttranslational event, is the essence of complex response to external and internal stimuli and cellular homeostasis [1,2]. Misbalance of protein phosphorylation is associated with cancer initiation, progression, metastasis, and drug resistance [2]. Protein phosphatases are key off switches of many different cellular pathways driven by about 520 protein phosphorylases, aka kinases, balancing dynamic process of protein phosphorylation [3].

Protein phosphatases are primarily classified based on the amino acid residue they dephosphorylate, and are therefore categorized into serine/threonine-, tyrosine-, or the dual specificity-phosphatases [4]. One of the most important classes of phosphatases is Ser/Thr phosphatases that encompass the phosphoprotein phosphatase (PPP) subclass (PP1, PP2A, PP2B, PP4, PP5, PP6 and PP7), metal-dependent protein phosphatase (PPM) subclass that are Mg^2+^/Mn^2+^ dependent, and the aspartate-based phosphatases [4]. Among these subclasses, PP2A enzymes are the most extensively studied proteins, especially in the context of cancer and brain tumors [5].

Serine/threonine protein phosphatase 2A (PP2A) is a ubiquitously expressed “holophosphatase [6]” and a major negative regulator of about 50 distinct proto-oncogenic signaling pathways [7]. Aside from its housekeeping roles in controlling proliferation, cell cycle, survival, apoptosis and DNA Damage Response (DDR), PP2A is also implicated in immune-checkpoint signaling, metabolism, cell-cell communication, cytoskeleton dynamics and neurotransmission, all contributing to cellular homeostasis. Similarly, stimuli such as hypoxia, drugs, and irradiation trigger increased PP2A expression to tackle stress-associated effectors [5]. Accordingly, PP2A has been used as a tool to modulate drug sensitivity in a handful of cancers, including GBM, IDH wt (hereafter referred to as “GBM”) [5,8,9,10,11,12].

PP2A has several subunits and isoforms with distinct functions [8,13]. The diversity of these subunits that constitutes each specific PP2A holoenzyme (aka PP2A heterotrimer), is already well-reviewed [13,14]. In short, the PP2A holoenzyme is comprised of a structural scaffold (A), and a catalytic (C) subunit, each having two isoforms (α and β). The immense diversity of PP2A holoenzymes is, however, determined by the third regulatory (B) subunit that has at least 21 different members categorized into four different subfamilies (B (B55/PR55), B′ (B56/PR61), B″ (PR48/72/130), B‴ (PR93/110/Striatin)) (Figure 1A) [3]. With almost no sequence similarities, these R subunits are responsible for substrate specificity and spatial localization of PP2A holoenzymes [9,14,15]. Furthermore, each of these R subunits possesses two to five isoforms and several splice variants [3]. This diversity makes up 80–100 putative PP2A holoenzymes. However, the abundance of multiple PP2A subunits is not ubiquitously alike and seems to be tissue and developmental stage-specific [16,17,18,19]. For instance, whilst PPP2R2A and PPP2R2D (B family) and PPP2R6B (B‴ family) are expressed rather ubiquitously across different human tissues, the expression of PPP2R2B and PPP2R2C (B family), PPP2R5B, PPP2R5D and PPP2R5E (B′ family), and PPP2R6A and PPP2R6C (B′′′ family) seem to be more exclusive to brain tissue [14,20,21,22]. Likewise, PPP2R5A and PPP2R5C are B′ family members that are more abundant in cardiac cells [23,24,25]. Regulatory (R) subunits compete with each other in binding to the AC core, forcing the other outcompeted R monomers into proteolysis; this process is tightly regulated [13]. Despite the relatively high sequence homology, PP2A isoforms can barely functionally compensate for each other. In this regard, it is shown that PPP2CA knockout mice could not survive, despite the abundant PPP2CB expression and 97% sequence identity [14]. Similarly, despite the fact that the structural subunits of Aα and Aβ are 85% identical, their affinity to the B and C subunits is different, making them functionally rather distinct [3]. The pleiotropy of PP2A is also determined, at least in part, by its subcellular localization. In this view, cytoplasmic PP2A regulates processes such as cellular growth, survival, and metabolism, whereas nuclear PP2A is mainly involved in cell cycle progression, regulation of DDR, and chromosomal stability. Finally, PP2A also localizes to the mitochondrial membrane, where it regulates apoptosis [26].

## 2. PP2A Regulation

### 2.1. PP2A Regulation in Normal Cells

PP2A expression is tightly regulated due to its significant contribution to more than 50% of mammalian Ser/Thr phosphorylation events [18]. Binding sites of several transcription factors with various functions have been identified in the proximal promoter of many of these PP2A subunits [11,16]. Moreover, epigenetic regulation plays an important role in PP2A expression by altering chromatin accessibility. Furthermore, it has been reported that PP2A subunits can be regulated by microRNAs [13,27]. PP2A regulation is mainly at translational and post-translational levels (Figure 1B). Importantly, endogenous PP2A binding partners can repress/activate certain PP2A holoenzymes via post-translational modification (L309 methylation and Y307/T304 phosphorylation of C-terminal of C subunit) or simply by abolishing holoenzyme assembly via imposing conformational changes [28]. In addition, Simian Virus 40 (SV40) small t antigens (ST) can bind to the PP2A structural core (Aα), disrupting its interaction with B subunits [29]. Schematic Figure 1B summarizes the most prevalent players of this tight regulatory network.

### 2.2. Somatic Alterations and Dysregulation of PP2A in Cancer

PP2A is most often inactivated in solid (lung, colorectal, glioma, ovarian, melanoma, among others) and hematopoietic cancers [14]. Therefore, the more classical opinion is that PP2A is a *bona fide* tumor suppressor gene in cancer, since it counteracts the aberrantly activated oncogenic kinases. On the other hand, there are a handful of studies suggesting that PP2A activation can also support tumorigenesis [20]. Interestingly, in each cancer type, different sets of PP2A subunits are dysregulated, highlighting the PP2A holoenzyme tissue and functional specificity in the context of cancer. Mutations are generally considered a less frequent cause of PP2A inactivation compared to dysregulation by transcription factors, aberrant epigenetic alternations and microRNA dysregulation [8,13]. Somatic PP2A mutations are usually heterozygous, creating haploinsufficiency [3]. PP2A missense mutations commonly occur in the binding domain of the structural (A) subunit to the B subunit, interfering with protein complex formation [30]. Nevertheless, the frequency of mutations in PP2A subunits varies considerably depending on the cancer type [31]. Most common cancer-associated PP2A dysregulation mechanisms involves post-translational modifications that are described in Section 2.1. Furthermore, several oncogenic microRNAs are able to target PP2A subunits for mRNA degradation and/or translational inhibition (Table 1 and Figure 1) [32]. For instance, miR-222 negatively regulates PPP2R2A in liver cancer, contributing to elevated Akt signaling [33]. Likewise, our group previously showed that miR-19b contributes to EGFR activation in non-small cell lung cancer (NSCLC) by targeting PPP2R5E [27]. Conversely, miR-374a-5p, miR-542-3p and miR-19b-3p can indirectly contribute to PP2A restoration by targeting MID1, an E3 ubiquitin ligase, driving PP2A degradation in Alzheimer’s and neuropathies [34]. A list of the most deregulated PP2A subunits together with the mechanism of deregulation for most common cancer types is provided in Table 1. It is worthy of note that PP2A dysregulation is generally associated with poor prognosis.

### 2.3. Somatic Alterations and Dysregulation of PP2A Subunits in Gliomas

PP2A is inactivated or dysregulated in approximately 60% of GBM cases [5]. Gene deletion, missense or nonsense mutation, and frameshifts are reported as PP2A-deactivating mechanisms in gliomas [5,14,60], however loss of heterozygosity (LOH) resulting in homozygous deletion of PP2A structural subunits (PPP2R1A and PPP2R1B) is less frequent than in other cancer types. Given the chromosomal location of the PPP2R1A gene (19q13.1-13.4) this scarcity is surprising, because the LOH of this locus is expected in 25% of GBM, and per definition, 100% of oligodendroglioma, IDH mutant and 1p/19q-codeleted tumors [61]. In addition, PP2A expression is reduced in 40% of glioma cases at the level of transcription, mainly due to the negative regulation of transcription factors [11,31]. Little is known about the role of microRNAs that regulate PP2A expression in gliomas. However, we showed that miR-19b, which is frequently deregulated in GBM, contributes to reduced PP2A expression and chemotherapy resistance (Kashani et al. [62], unpublished results). Likewise, miR-181b-mediated PP2A repression in GBM cells contributes to increased GBM invasion through the deactivation of transcription factor-specific protein-1 (SP-1), a major target of the JNK pathway [63].

Irrespective of the mechanism of inactivation, dysregulation of PP2A structural subunits is per se oncogenic as it unbalances the abundance of other PP2A subunits owing to the failure of structural crescent hook-shaped protein [64] (due to reduced expression and/or mutations resulting in a misfolded protein [31]) to congregate C and R subunits in form of a fully functional PP2A holoenzyme. In this view, despite global PP2A downregulation, excessive free unincorporated C subunits is reported in GBM cases with mutations in one of the two isoforms of A subunit [31]. The exact consequence of this surplus of C subunit is not yet fully understood. However, an interesting study by Jiang et.al. suggests the role of mammalian α4 protein as a monomeric C subunit stabilizer to counteract the uncontrolled phosphatase activity of the monomeric C subunit while recycling and feeding it into non-canonical PP2A signaling pathways, providing adaptation to stress [17,65]. Mechanisms of dysregulation of PP2A subunits implicated in gliogenesis are listed in Table 1.

In GBM, we utilized the TCGA dataset to calculate the hazard ratio based on the cox-regression model comprehensively for all the PP2A subunits and PP2A binding proteins for which TCGA data (mRNA) was available. Whilst expression of the majority of PP2A holoenzymes was associated with a tendency towards reduced hazard ratio (HR), only PPP2R5D and PPP2R3A downregulation was significantly associated with worse survival in a multi-variant model (HR = 0.43 and 0.4, respectively) (Figure 2A). Interestingly, patients with higher levels of PPP2R1B and PPP2CA mRNAs have significantly worse survival (HR = 2.04 and 3.03, respectively), again highlighting the pleiotropic role of PP2A in GBM and supporting the disapproval of the classical hypothesis that PP2A is a *bona fide* tumor suppressor gene. Importantly, the expression of none of these genes was associated with age, IDH1 mutation status, gender, tumor mutational burden and other confounder factors as confirmed by multivariate analysis (Spearman *p*-value > 0.05). In addition of being a favorable prognostic marker, analysis of survival data using an ROC plotter indicated that patients with high level expression of PPP2R3A are associated with a better response of GBM patients to temozolomide (TMZ) (Figure 2B). Indeed, PPP2R3A expression reduced the hazard ratio by a factor of 0.4 (Figure 2A), suggesting that it can potentially be used as a predictive marker to TMZ response.

### 2.4. Role of PP2A Inhibiting Proteins (PAIPs)

As described earlier, PP2A is regulated at least in part by endogenous PP2A inhibiting proteins such as CIP2A, PPME1, PHAP1 (ANP32A), SET, SETBP1, among others, that inhibit the PP2A heterotrimer based on the cellular context and the spatiotemporal regulation. These proteins are frequently upregulated in cancer, suggesting a common mechanism that contributes to PP2A downregulation [19]. 

The overexpression of PAIPs has been reported in many cancers (Table 1), despite the fact that the underlying molecular mechanism is not deciphered. However, tissue-specific microRNAs such as miR-375 and miR-383 in HPV-associated cancers and lung adenocarcinomas, respectively [66,67], and the activation of upstream signaling pathways such as EGFR-MEK-Erk [68], and Scr/Erk [69] may contribute to the upregulation of CIP2A and other PP2A inhibiting proteins.

The cancerous inhibitor of PP2A (CIP2A) inhibits PP2A and is considered as an oncogene in several malignancies including CRC [3,28]. CIP2A can directly bind to and block B′ (B56)-containing PP2A complexes [61]. Furthermore, CIP2A negatively affects PP2A-mediated c-MYC proteolytic degradation, which is associated with poor prognosis and hypoxia adaptation [28,70,71,72,73,74,75,76,77,78].

SET nuclear proto-oncogene (I_2_PP2A) is another potent endogenous PP2A inhibitor that is considered an oncogene in leukemia (AML, CLL & CML), lymphoma, CRC, breast, and lung cancer [9,10,18,79,80,81]. The enhanced activity or overexpression of SET is mediated by translocations resulting in gene fusions [82,83,84] and post-translational modifications such as phosphorylation at Ser9 and its subcellular localization [3], and is frequently associated with poor prognosis.

PME1 (protein phosphatase methyl esterase 1) is a methyl esterase protein that inhibits selectively those PP2A holoenzymes that contain the AC subunit. It acts as a specific demethylase of amino acid residue L309 at the C-terminus of PPP2CA (Figure 1B) [85,86]. This post-translational modification leads to denaturation of the active site and interferes with the binding of PPP2CA to B55 subunits, contributing to PP2A inactivation [3,14]. Interestingly, methylation of C-terminal tail of PPP2CA is mediated by a SAM-dependent methyl transferase (LCMT-1), which is pivotal for assembly with B subunits, but not B’, B’’ and B’’’ subunits [3,87]. PTPA (PTPase activator, PPP2R4) is a PP2A activator that acts as a PP2A chaperon facilitating post-translational modification of the correctly folded AC core [88,89]. Specifically, PTPA keeps the PPP2CA subunit in a methylated, active state by preventing PPME1 association [90]. Importantly, PPP2CA can undergo phosphorylation at Y307 and T304 residues and this can prevent proper C subunit methylation, limiting interactions with B subunits. This latter mechanism interferes mainly with PPP2R5D and PPP2R2A heterotrimer formation [3,91].

PHAP1 (I1PP2A/ANP32A) is another PP2A endogenous modulator. It is overexpressed in many cancer types, including CRC and GBM, contributing in transformation through PP2A inactivation, leading to increased Akt signaling [92,93]. PHAP1-mediated PP2A inhibition in endothelial (HUVEC) cells is shown to be nullified by sphingosines, leading to PP2A reactivation [94].

### 2.5. Role of PP2A Inhibitory Proteins in Gliomas

CIP2A and SET overexpression are major mechanisms of PP2A inhibition in GBM [5,95]. CIP2A overexpression in particular is reported in the classical subtype of GBM with EGFR amplification [61]. Similarly, the expression of PPME1 is detected in at least 50% of astrocytoma, IDH mutant [86] and is associated with progression to higher grade and resistance to multi-kinases inhibitors (like PI3K inhibitor, LY29644 and multi-RTK inhibitor, sunitinib) [85]. Furthermore, ARPP19 (Cyclic AMP-regulated phosphoprotein 19) is another brain-enriched PP2A inhibitor and substrate of protein kinase A that is upregulated in higher-grade gliomas [96].

## 3. PP2A as Modulator of Oncogenic Signaling Pathways

Depending on the composition of the heterotrimer complex, PP2A is able to modulate diverse signaling pathways during tumor initiation and progression. PP2A also modulates pathways conferring resistance to conventional therapies. In addition, PP2A is responsible for the high plasticity of tumor cells in GBM [97,98,99]. In this chapter we describe the role of PP2A in modulating oncogenic signaling pathways. Figure 3 summarizes main complex signaling pathways affected by PP2A. Majority of these pathways including PI3K, EGFR, Myc, WNT, JAK/STAT and NF-κB pathways are frequently dysregulated in GBM [100,101].

Protein dephosphorylation by phosphatases can cause the activation or deactivation of the substrate. In the following, we refer to the inhibitory effect of PP2A-mediated dephosphorylation as ID (inhibiting dephosphorylation) and the activating effect of PP2A-mediated dephosphorylation as AD (activating dephosphorylation).

### 3.1. PI3 Kinase/Akt Signaling

The PI3K/Akt/mTOR pathway is activated by receptor tyrosine kinases and controls important cellular processes such as proliferation, migration, apoptosis and angiogenesis [102]. In solid cancers, hyperactivated PI3K/Akt/mTOR signaling is either due to activating mutations in EGFR, ERBB2, PIK3CA and Akt1, or loss of function of tumor suppressor genes such as PTEN [102]. Sustained phosphorylation of PIK3CA, Akt and downstream effectors of mTOR including p70S6K or 4EBP1, is also mediated by the downregulation of B subunits including PPP2R5B, PPP2R5C and PPP2R2B [13,103,104].

### 3.2. MAPK Pathway (Ras/Raf/MEK/Erk)

Like the PI3K/Akt/mTOR pathway, the RAS/MAPK pathway is also activated by receptor tyrosine kinase signaling and mediates proliferation, migration, apoptosis and angiogenesis [105]. In solid cancer, the constitutive activity of this pathway is induced by mutations in KRAS, NRAS and BRAF [105]. Reduced PP2A activity can further enhance the activity of the RAS/MAPK pathway. PP2A directly deactivates Erk2, MEK1, Ras and Raf [106]. Notably, PPP2R5C is important for pErk inactivation. Conversely, pErk can deactivate PPP2R5C by Ser337 phosphorylation, leading to PP2A disassembly and inactivation and thereby forming part of a negative feedback loop contributing to fine tuning modulation of Erk signaling in normal tissues [106,107].

### 3.3. C-MYC

The proto-oncogene c-MYC is a transcription factor that controls many cellular processes including cell cycle progression [108]. c-MYC is commonly overexpressed in many human malignancies, and contributes to cellular transformation [108]. PP2A (particularly, PPP2R5A [109]) dephosphorylates c-MYC on Ser62 (conserved residue responsible for c-MYC stabilization), thereby targeting c-MYC for ubiquitination and proteasomal degradation. Interestingly, CIP2A-mediated PP2A inhibition or pharmacological inhibition of PP2A using okadaic acid (OA) increases c-MYC levels and contributes to sensitivity to chemotherapies such as paclitaxel, doxorubicin and temozolomide in melanoma cells [9,110]. This regulation is also specifically important due to the role and tight crosstalk of c-MYC protein with the RAS-RAF-MEK-Erk protein kinase pathway [111].

### 3.4. Wnt/β-Catenin Signaling

The role of PP2A on the Wnt signaling pathway is rather controversial: in normal cells with a low activity of Wnt signaling, PP2A protects cells against excess Wnt pathway activation: PP2A is trapped in the complex together with APC, Axin, GSK-3b, and DVL, and therefore fails to dephosphorylate β-catenin. Phosphorylated β-catenin will be targeted for proteasomal degradation, resulting in reduced Wnt signaling [20]. However, in CRC, pancreatic cancer and GBM, PP2A complexes containing PPP2CA and PPP2R2A form part of a positive feedback loop contributing to its activation [9,20]. Accordingly, PPP2R2A expression is associated with poor survival in pancreatic cancer [20]. This is in part due to PP2A-mediated GSK3B kinase dephosphorylation (AD), which is reversed by PP2A inhibition in glioblastoma stem cells (GSCs) [112].

### 3.5. JAK/STAT Pathway

The Janus kinase-signal transducer and activator of transcription (JAK-STAT) pathway plays a major role in cytokine signaling of immune cells. IL-2 is a potent inducer of JAK/STAT signaling in lymphocytic malignancies [9,113]. The PP2A holoenzyme interferes with JAK/STAT signaling by targeting JAK3 and STAT5 effector proteins [113].

### 3.6. Src Pathway

Non-receptor tyrosine kinase Src is a master regulator of cell proliferation [114]. Hyperactive Src elicited by reduced PP2A activity is a driver of cellular transformation and carcinogenesis, and contributes to extrinsic apoptosis resistance [9]. Src inhibition is mediated by PPP2R2C in a cell-type specific manner [115], as Scr is a PP2A substrate in osteosarcoma but not prostate cancer cells [7].

### 3.7. NF-κB Pathway

NF-κB has a dual role in pro-survival signaling and extrinsic apoptosis [116]. Upon stimulation by external signals or stress, IKK is activated and phosphorylates IκB (inhibitory protein), targeting it to ubiquitin-mediated protein degradation. As a result, NF-κB is released and translocates into the nucleus where it transactivates multiple genes involved in proliferation, invasion and apoptosis [117]. PP2CA interferes with NF-κB signaling by IKK dephosphorylation. Depending on the cellular context, PP2A inhibitors can either enhance apoptosis or pro-survival signaling [118].

### 3.8. C-Jun/JNK Pathway

C-Jun N-terminal Kinase (JNK) is a stress-activated kinase of the MAPK family, which controls proliferation, apoptosis, DNA repair, and metabolism, and depending on the cancer system can act either as an oncogene or tumor suppressor gene [119]. It has been reported that PP2A heterotrimers containing the PPP2CA subunit are responsible for the dephosphorylation of JNK (ID) [120]. 

### 3.9. Hippo Pathway

The Hippo pathway is implicated in tissue regeneration, tumorigenesis, and immune responses [121]. PPP2R6B (STRN3)-containing PP2A complexes can inactivate the Hippo pathway by dephosphorylation of pMST1/2 kinases, which in turn lead to YAP activation and tumorigenesis [6]. Enhanced PPP2R6B expression is associated with YAP overexpression, poorly differentiated histology and the poor prognosis of gastric cancer [6].

## 4. Role of PP2A in Cellular Processes

In this chapter, we describe the role of PP2A in modulating different cellular processes, focusing on main cancer hallmarks. The schematic in Figure 4 summarizes the role of PP2A in cell cycle progression and DNA repair, and Figure 5 summarizes the role in modulating various cancer associated cellular processes. These processes may also contribute to the TMZ/RT response of GBM patients.

### 4.1. Cell Cycle Progression

PP2A controls cell cycle progression at G1, G2/M and during the M phase by targeting cell cycle regulatory proteins [19,122].

G1 to S transition is regulated by cell cycle regulators such as p27. PPP2R5C accumulates in the nucleus during the G1 and S phase of the cell cycle and negatively regulates cell cycle progression through p27-dephosphorylation (AD) [123]. Cell cycle arrest in G1 can also be induced by PP2A-mediated dephosphorylation of RB1 [124].

Mitotic entry requires activation of CDK1 and the concomitant inhibition of PP2A, which dephosphorylates CDK1, resulting in its inhibition. PP2A inactivation is mediated by Greatwall kinase (GWL, Mastl) [9,125,126] leading to further inhibition of PP2A via the activation of ARPP19, a potent PP2A inhibitor, allowing cells to progress through mitosis. Conversely, following DNA damage, PP2A inhibits PLK-1 (Polo-like Kinase-1) (ID), thereby triggering G2/M cell cycle arrest and activation of DNA damage response [127]. PP2A can also prevent mitosis entry by deactivating Aurora B and PIK1 kinases (ID). This leads to prolonged mitotic spindle checkpoint response, ensuring proper spatiotemporal microtubule attachment to the kinetochores as a surveillance mechanism before entry into the M phase [9,128]. Accordingly, premature mitosis entry has been reported in yeast and mammalian cells with dysregulated PP2A expression [13].

During mitosis, PP2A is recruited to the centrosomes where it generates a complex with SGO1 protein and contributes to centromeric cohesion and prevents aneuploidy and tumorigenesis [129].

Following mitosis completion, cells need to ensure timely mitosis exit and post-mitotic interphase assembly through dephosphorylation of CDK1, CDK1 substrates (such as Histone H1), and Cyclin E kinases. This process is mediated by the PP2A holoenzyme containing the PPP2R2A subunit [130]. However, this is corroborated by the ability of PPP2R2A to dephosphorylate and activate Wee1 and Myt1 (AD), which in turn lead to more efficient CDK1 inhibition [9,19]. Interestingly, this complex can also trigger dephosphorylation (ID), nullifying the inhibiting effects of ARPP19, all of which contribute to a successful mitosis exit [13].

In GBM stem cells, PP2A activity is enhanced by hypoxic conditions inducing G1 cell cycle arrest, tumor cell dormancy, and TMZ resistance [131], but the underlying mechanism of G1 arrest has not been resolved. 

### 4.2. DNA Damage Response and DNA Repair

DNA damage-induced cell cycle arrest is orchestrated by the time-dependent and sequential Ser/Thr phosphorylation of DDR kinases [10]. Phosphatases are responsible for termination of the DDR pathway following DNA repair, allowing cells to re-enter cell cycle progression. The role of PP2A in mediating DNA damage is mainly attributed to targeting pATM [9]. In this view, it is reported that PP2A deficient cells are arrested in G1/S due to ATM and CHK2 aberrant activation and hyperphosphorylation [10]. Sustained p-ATM activity can trigger phosphorylation of PPP2R5C complexes at Ser510, leading to PPP2R5C-containing PP2A complex formation, which contributes to p53 activation [128,132].

Other than ATM and CHK2, PP2A also cooperates with TORC1 and Irc21 (cytochrome b5, implicated in Ceramide synthesis) to attenuate ATR-mediated DDR [10]. This is further fortified by the observation that PP2A attenuated cells are also compromised in homologue recombination (HR) repair with marked downregulation of HR components such as RAD51 and BRCA1 [10].

One of the best characterized roles of PP2A in the context of DDR is the regulation of G2/M cell cycle transition. PP2A (particularly, PPP2R5D [133]) dephosphorylates and thereby inactivates Cdc25. Cdc25 is a protein phosphatase that dephosphorytes DDR kinases ATM, ATR, DNA-PK and their downstream phosphokinases, CHK1 and CHK2 enabling cell cycle progression [9,10]. In conclusion, by targeting Cdc25, PP2A blocks cell cycle progression in G2/M in response to DNA damage [134].

Other than DNA damage response, PP2A also plays an important role in DNA repair. This includes the homologous recombination (HR) pathway, in which PP2A-mediated deactivation of important DNA repair proteins such as RPA32 that is necessary for DNA repair completion [135]. Ambjorn et.al reported that a complex of the BRCA2 and B56 family is required for proper RAD51 foci formation at the site of the damage and successful HR [136]. Likewise, PP2A removes γH2AX from chromatin proximal to the break site, facilitating DNA repair. By deactivating DNA-PK, Ku70 and Ku80, PP2A is also implicated in Non-Homologous End Joining (NHEJ) DNA repair process [10]. 

TMZ in combination with radiotherapy is standard-of-care for GBM patients [137]. TMZ induces DNA adducts such as O^6^MeG, which after incorporation into the replicating DNA form mispairing with thymine leading to futile cycles of mismatch repair (MMR) response. DNA double-strand breaks eventually occur in response to this erroneous repair process, which, if not repaired, leads to cytotoxicity. However, DNA repair mechanisms such as O^6^-methylguanine-DNA methyl transferase (MGMT)-mediated repair, nucleotide excision repair (NER), base excision repair (BER), non-homologous end joining (NHEJ) and homologous recombination (HR) repairs nullify these cytotoxic lesions, contributing to TMZ resistance [138]. Although a direct link between PP2A activity and TMZ resistance has not yet been established, knocking down ATM in GBM cells abrogated BER and promoted therapeutic resistance to alkylating agents [139]. Thus, it is conceivable that PP2A is implicated in TMZ resistance by dephosphorylating ATM. PP2A also dephosphorylates p53, MDM2, p38, CHEK1 and CHEK2, thereby interfering with checkpoint responses. The activation of anti-apoptotic pathways, autophagy and senescence are survival strategies of GBM cells to cope with cytotoxic effects elicited by TMZ treatment [140]. 

In conclusion, PP2A restoration might be a useful tool to overcome the TMZ resistance of GBM. More in depth studies are needed to determine when and how to utilize PP2A modulation to overcome TMZ resistance in gliomas.

### 4.3. Apoptosis

PP2A localization at the mitochondrial outer membrane suggest a role in regulating apoptosis [19]. PP2A can either induce (more commonly) or inhibit (rather non-canonically) apoptosis. 

The PP2A role in respect to apoptosis is driven at least by five distinct mechanisms [9]:(1)PP2A induces apoptosis by interfering with the PI3K/Akt prosurvival signaling pathway.(2)It can directly target pro- and anti-apoptotic proteins. It induces apoptosis by targeting the anti-apoptotic protein, BCL2 [20] (ID), and pro-apoptotic molecules, Bad (AD) and Bim (AD) [19]. Conversely, PP2A confers apoptosis resistance by targeting the Caspase 2 inhibitor, CaMKII (AD).(3)PP2A induces apoptosis by targeting p53. Following DDR, PP2A contributes to p53-mediated apoptosis by dephosphorylating p53 at Thr55 (mediated by PPP2R5C [30,132]) (AD) (see Figure 3). After DNA is repaired, PP2A dephosphorylates p53 at Ser37 (AD), thereby reducing p53 transcriptional activity [10]. PP2A can also induce apoptosis by targeting the negative p53 regulators, p-MDM2 and p-EDD (ID), thereby enhancing p53’s protein stability [128,141].(4)PP2A confers resistance to TRAIL (TNF-Related Apoptosis Inducing Ligand-mediated)-induced cell death by dephosphorylation of the non-receptor tyrosine kinase Scr, a mediator of the TRAIL signaling, and thereby contributes to therapy resistance and metastasis of breast cancer [9].(5)PP2A is also implicated in ER stress-induced apoptosis through BIM activation (AD) and BCL2 inhibition (ID), thereby contributing to increased BCL2/p53 interaction and ER apoptosis induction [142].

### 4.4. Epithelial to Mesenchymal Transition

Epithelial to mesenchymal transition (EMT) is an important mechanism facilitating cancer cell metastasis. The loss of the adherens junction is a hallmark for EMTness and is mediated mainly through E-cadherin [143]. PP2A (PP2A-AC) is localized at the cell-to-cell adherens junctions in complex with E-cadherin, supporting epithelial/polygonal morphology and preventing mesenchymal ization [144]. Furthermore, by deactivating (ID) matrix metalloproteinase (MMPs), PP2A counteracts ECM degradation and, subsequently, EMT. On the other hand, VEGF-induced YAP activation is mediated thorough a PP2A dephosphorylation event (AD) that contributes to invasiveness and angiogenesis, supporting tumor metastasis [145]. YAP activation is further fortified by PP2A-mediated MST1/2 deactivation and Hippo pathway turn off (see Section 3.9). 

### 4.5. Senescence

Oncogene-induced senescence is a mechanism to cope with elevated ROS production in highly proliferating cancer cells. PP2A is directly involved in redox regulation and cellular protection against elevated ROS and oxidative stress [10,146]. In response to elevated ROS in HUVEC cells, DNA damage response is constantly induced. PP2A mediates translocation of nucleophosmin (NPM) to the nucleus, preventing prolonged γH2AX foci formation, and terminal senescence, assuring functional endothelial cells despite the constant oxidative stress. 

Furthermore, PPP2R5E-MCTL1 (microtubule cross-linking facor-1) interaction enhances the stability of the microtubules, preventing Golgi fragmentation and microtubule disruption that are characteristics of senescent cells [147,148].

On the other hand, PPP2R5A downregulation contributes to increased c-MYC-induced senescence in melanomas [47]. Furthermore, the central “energy-sensing” kinase, AMP kinase (AMPK), blocks mTORC1, and thereby inhibits fatty acid and lipid synthesis, increases p53-stablization and autophagic cell death, and is considered as a tumor suppressor gene in many cancers including CRC [149]. PPP2R2D can inhibit AMPK and rewire the lipid metabolism by preventing AMPK-mediated shut down of anabolic pathways. This fuels cancer progression, suggesting PP2A inhibition as a therapeutic approach to combat metabolic disorders and cancer progression [150]. Many of these increased synthesised fatty acids are senescence-associated lipid changes that give rise to therapy resistant persister cancer cells [151].

Protein arginine methyltransferase 5 (PRMT5) is overexpressed in stem-like GBM, but depletion of PRMT5 results in enhanced senescence [152]. Interestingly, PRMT5 depletion also increased PP2A activity by a yet not defined mechanism. Senescence is not an irreversible phenotype, since therapy-induced senescent cells maintain their capacity to reenter the cell cycle and contribute to tumor recurrence. Intriguingly, PRMT5-depleted senescent GBM progressed to necroptosis when treated with LB100, suggesting a role of PP2A in maintaining the reversible state of senescence. Thus, the dual inhibition of PRMT5 and PP2A can be a potential new therapeutic strategy for the treatment of GBM [152].

### 4.6. Autophagy

Emerging evidence supports a role for autophagy in therapy resistance by inducing cell dormancy and persister tumor cell formation [153]. Beclin1 and ULK complex formation are indispensable for (macro)autophagy induction and the generation of autophagosomes. PPP2R2A can interfere with autophagy by targeting Beclin1 and ULK1 proteins [154]. PP2A can also target AMPK, a central regulator of autophagy (see Section 4.5).

### 4.7. Hypoxia and Angiogenesis

PP2A is upregulated in response to hypoxic stress [5]. PP2A contributes to HIF1-α downregulation and the inhibition of vascularization (elevated VEGF levels) in order to reduce the metabolic demand of GBM and pancreatic cancer cells [131,155]. This may have interesting therapeutic implications: pancreatic cancer cells are poorly perfused and therefore respond poorly to therapy owing to reduced local drug concentrations in the tumor. PP2A inhibition may improve therapy response by means of increased VEGF-mediated angiogenesis [10]. On the other hand, hypoxic conditions enhance c-MYC signaling via HIF2-α expression, promoting renal cell carcinoma [78]. C-MYC destabilization by PP2A inhibition could be exploited therapeutically to interfere with the adaptation of tumor cells to hypoxic conditions [78].

### 4.8. Stemness and Dedifferentiation

PI3K/Akt, MAPK and Wnt/β-catenin signaling pathways elicit essential regulatory functions on cancer stemness [143]. The role of PP2A in modulating these pathways was discussed in a previous chapter of this review. A common kinase involved in these pathways is moonlighting kinase glycogen synthase kinase-3 (GSK-3). The role of GSK-3 in EMT, cancer development and metastasis is discussed elsewhere [156]. GSK-3 itself is also a target of PP2A. PP2A restoration has shown promise in inducing the differentiation of BCR/ABL CML cells [84]. In gliomas, it is proposed that PP2A contributes to the stemness of gliomas. The proposed mechanism may be explained by the finding that PP2A deactivates PI3K/Akt signaling and thereby interferes with N-COR cytoplasmic degradation [9]. In nestin-positive U87 GBM cells, PP2A inhibition induces astroglial differentiation [9]. Furthermore, PP2A holoenzymes containing the β isoform of A subunit (PP2A-Aβ) lead to the dephosphorylation of oncogenic Ras-GTPase RalA and tumor cell differentiation [157].

### 4.9. Pro-Inflammatory Signals

PP2A counteracts aberrant inflammatory responses by targeting inflammatory mediators and matrix metalloproteinase (MMP). Indeed, PP2A has been identified as a new player in chronic obstructive pulmonary disease (COPD) and asthma, and contributes to lung cancer initiation and progression by creating a tumor-promoting inflammatory microenvironment [158,159]. The role of PP2A in mediating a favorable microenvironment is in part due to its ability to target MMPs. In particular, loss of PP2A activity (mainly through CIP2A overexpression) causes increased MMP1 and MMP9 secretion, promoting lung inflammation and tumorigenesis [160]. Increased MMP-9 can also induce activation of the pro-angiogenetic ligand VEGF from its latent form in the extracellular matrix, enhancing tumor neoangiogenesis. Furthermore, PPP2R5A localization to the heart myofilaments triggers dephosphorylation of contractile proteins and regulation of the Ca^2+^ flux. Thus, PP2A may play an important role in coordinated cardiac contractility and inflammatory response [161].

Immune checkpoint inhibitors, such as PD1 blockade, failed to show clinical benefit for the treatment of GBM. Combination therapy using LB-100 and PD1 blockade significantly improved survival compared with monotherapy alone in a murine glioma model [162], demonstrating the importance of PP2A as an immune modulator. 

### 4.10. Phospho-Tau in CNS Disorders and Cancer

One of the important PP2A targets in the context of neurological disorders is protein tau. PP2A accounts for approximately 71% of the total tau phosphatase activity of the human brain [3,9]. Hyper-phosphorylated tau protein (aka taupathy) is the root cause of neurofibrillary tangles that contribute to the pathology of Alzheimer’s disease, as it causes disruption in the assembly of affected neuron cells [163,164]. Tau hyperphosphorylation is due to PP2A dephosphorylation at Y307, rendering PP2A inactive [165]. In contrast, PP2A-C subunit methylation is pivotal in keeping Tau proteins hypo phosphorylated, enabling direct co-localization of Tau with methylated PPP2R2A subunits, preventing Alzheimer’s disease onset [166]. Interestingly, the high expression of active Tau in the tumor is associated with reduced glioma growth and the prolonged survival of glioma patients [167]. Tau is also involved in resistance to taxanes in various tumors such as breast, ovarian and gastric carcinomas. The proposed resistance mechanism is explained by the fact that taxanes and Tau compete for the same binding sites on microtubules [167].

## 5. Clinical Trials Using Activators or Inhibitors of PP2A

A great effort was spent on restoring PP2A activity in pancreatic cancer and AML by means of genetic approaches aiming to knock down the PP2A endogenous inhibitors, SET and CIP2A, as well as pharmacological approaches using PP2A activating drugs (PADs) or small-molecule activators of PP2A (SMAPs) to prevent tumor growth and overcome therapy resistance [5,10,168]. Furthermore, PP2A restoration has also shown promise in reducing GBM cell populations with EMT-like phenotype and angiogenesis, while inducing cell death [169]. The most frequent drug used in this context is FTY720 (fingolimod) from Novartis that was initially used as a drug for the treatment of multiple sclerosis. Grossman et al. examined the feasibility of combining Fingolimod to the standard-of-care for newly diagnosed GBM patients in a single arm trial [170]. The results of this trial highlighted important safety concerns with this combination, including prolonged lymphopenia [170]. It is reported that FTY720 can indirectly activate PP2A by blocking SET binding [171]. In contrast, tricyclics derivatives such as phenothiazine bind to PP2A-Aα-containing heterotrimers, imposing activating conformational changes [172]. A full list of PP2A activating drugs used for the treatment of cancer is provided in Table 2.

However, owing to the dual role of PP2A as an oncogene and tumor suppressor gene, small molecule PP2A inhibitors were also tested for the treatment of cancer [173]. Indeed, PP2A inhibitors forced quiescent hypoxic cancer cells to enter mitosis (as Cdc25 was unleashed), rendering them susceptible to chemo-radiation therapy regimens that are tailor-made for highly proliferating cancer cells [174]. For example, knocking down PP2A or PP2A inhibition by okadaic acid (OA) contributes to c-MYC stabilization, aberrant proliferation, and increased sensitivity to paclitaxel, doxorubicin and temozolomide in melanoma cells [111]. Furthermore, since PP2A-mediated γH2AX removal facilitates DNA repair, it has been proposed that PP2A inhibition can hamper DNA repair, thereby sensitizing cells to DNA damaging agents [10]. It has been speculated that OA treatment in GBM cells drives damaged cells into a mitotic catastrophe [9], but PP2A inhibitors never progressed beyond phase I clinical trials. Some reports have cast doubt that PP2A inhibition causes cell-cell junction instability that might compromise blood brain barrier integrity [175]. Specifically, there are no ongoing clinical trials utilizing PP2A inhibitors for brain malignancies [175]. More studies are needed to address these challenges in a systematic manner.
cancers-14-05227-t002_Table 2Table 2PP2A activating/inhibiting drugs as a therapeutic anti-cancer approach.CompoundTargetModelsEffectCancer Therapy Realm StatusMolecule Class/Derivate ofOkadaic acidPP1, PP2A & PP4Various cancer cell linesReduced resistance to chemo and radiation therapy (RT)Pre-clinicalNatural product (dinoflagellates): toxin complex polyether fatty acid [176]Calyculin A *^2^PP1, PP2A & PP4Various cancer cell linesReduced resistance to chemo therapy and RT [177,178,179]Pre-clinicalNatural product (marine sponge extract)Microcystins *^1^ [63,180]PP1, PP2A & PP4Colorectal, liver, and prostate cancers [181]Inhibiting the catalytic activity of PP2A [182]Pre-clinicalNatural product (cyanophyte) cyclic peptide inhibitors [182]TautomycinPP1 (high affinity) & PP2A (lower affinity)medullary thyroid cancer cells [183]inhibition of GSK-3β Pre-clinicalantifungal antibiotic isolated from the bacterium Streptomyces verticillatusCantharidin *^1^ [184]P1, PP2A (most potent [118]) & PP4Lung, bladder and pancreatic cancerInduces ROS production and DNA damage, leading to apoptosis [185]Phase 1: (associated with urologic toxicity) [186]Natural product (Blister Beetle) [9]FostriecinPP2A & PP4 (potent), PP1 (weak) [187]Various cancer cell linesInhibits mitotic entry checkpoint through PP2A inhibition [187]Phase 1: toxicity and instability Natural product (antibiotic produced by Streptomyces pulveraceus)Dragmacidin *^2^ [188]PP1 & PP2A inhibitorLung, Colon and Breast cancermitotic arrest at metaphase [189]Pre-clinicalNatural product isolated from the Tanzanian sponge *Dragmacidon* sp.LB100 *^1^ [190] (& LB102)potent and selective inhibitory activity against PP2A (PPP2CA)GBM (+TMZ) [191]potent chemo- and RT-sensitizing propertiesby triggering mitotic catastrophephase-1competitive small molecule inhibitor of PP2APheochromocytoma (+TMZ) [127]Growth suppression and regression of metastasisSarcoma (+Doxorubicin) [9]
Hepatocellular carcinoma (+Doxorubicin/+Cisplatin)
Ovarian cancer [192]Cisplatin SensitivityPancreatic cancerDoxorubicin Sensitivity [155], RT sensitivity [193]Asprin *^1^ [194,195]Phosphorylative PP2A deactivationCRCsuppresses Wnt signalingPre-clinicalSynthetic organic compoundCurcumin *^2^ [196]PP2A Promoting GBM apoptosisActivation of mitogen-activated protein kinases [197]Pre-clinicalPlant-derived polyphenolAnisomycin *^2^PP2APromoting GBM apoptosis [198]down-regulation of PP2A catalytic subunitPre-clinicalAntibiotic retrieved from the bacteria Streptomyces griseolusPerphenazine *^1^(aka Phenothiazine)Induced PP2A holoenzyme activity by binding to the PP2A-A subunits KRAS-driven NSCLC [168]Sensitizer to MK2206 (Akt inhibitor) & AZD6244 (MEK inhibitor) [168]Pre-clinical (in vitro & in vivo)A tricyclic neuroleptic SMAP (re-engineered version of tricyclic sulphonamide)COG449 (OP449)Increased PP2A cellular activity and decreased Mcl-1 expressioncytotoxicity for CLL and NHL cells in vitroinduced cell death in a dose-dependent manner, sensitizes myeloid leukaemia cells [199]Pre-clinicalSmall peptide binding to SET (a dimerized derivative of COG112) [200]FTY720 *^1^ [201] (Fingolimod,Gilenya [Novartis] [5])Increased PP2A activity by disrupting PP2A/SET activity & E2F activation (AD) [168]Apoptosis induction [202,203,204](specific PP2A activator [18,204])Leukemia (CML) [205,206]induces apoptosis through the inactivation of BCR-ABL1 and Erk signalingPre-clinicalsphingosine analog, derived from fungal metabolite, immunosuppressantcolon cancer [207,208]
Pre-clinicalnon-small cell lung cancer [171]
Pre-clinicalbreast cancer [208]Trastuzumab and Lapatinib resistance [209]Phase1, ongoing [210]hepatocellular carcinoma [211,212]
Pre-clinicalprostate cancer [213]
Pre-clinicalGBM
Phase 1, 2015-17, failed due to induced lymphopenia [214]ApoE *^2^ [215] (Apolipoprotein E) and apoE-mimetic peptides(OP_449_ [199] and COG_112_ [216])SET inhibitorPrimary mouse peritoneal macrophages [217], Prostate cancer, CML and AML [207]Decrease activation of inflammatory signal; Inhibit tumor growthPre-clinical (in vitro and in vivo)ApoE and apoE-mimetic peptides (Synthetic small molecule inhibitor)TGI1002 [218]SET inhibitorCMLcombats multi-drug resistancePre-clinical
Sodium Selenite *^2^ [219,220]PP2A and PTEN activatorProstate cancer (increased apoptosis and ROS production)
Pre-clinical
EMQA (TD19) [221]SET inhibitorNSCLCPaclitaxel sensitivityPre-clinicalSynthetic small molecule inhibitorBortezomib *^2^CIP2A inhibitorTNBC (triple negative breast cancer) [75]Apoptosis inductionPre-clinical (in vitro and in vivo)Synthetic proteasome 20S inhibitor [75]head and neck squamous cell carcinoma cells [76]Apoptosis induction, pAkt reductionPre-clinical (in vitro and in vivo)Forskolin *^1^ (Colforsin) [222]SET inhibitorAML [84]abolished BCR/ABL phosphorylation and activationPre-clinicala diterpene derived from the roots of Coleus forskohlii [19]Chloroethyl Nitrosourea *^1^ (CENU)PP2A activator (via methylation [222])MelanomaGrowth inhibitionPre-clinicalalkylating agent [211]α-tocopheryl succinate *^2^ [223] (α-TOS)PP2A activator [224]Hematopoetic cancer cell lines & CRCApoptosis induction Pre-clinicalVitamin E analog [225]Carnosic acid *^1^PP2A activationProstate cancer [226]Inhibition of Akt/IKK/NF-κB signalingPre-clinicalPolyphenolic diterpene, isolated from the Rosemarinus officinalis [226]PP2A deactivation (via C subunit demethylation by PME1)Type II diabetes [227]Triggers insulin sensitivityPre-clinicalMethylprednisolone *^1^ [228]PP2A activation (mainly the R subunits)LeukemiaDifferentiation Induction Pre-clinicalSteroid hormone Ceramide *^1^ [229,230]PP2A activation (direct and SET-mediated)Prostate cancerApoptosis induction, p27 activationPre-clinicalSphingosine and fatty acid molecule (sphingolipid)Cucurbitacins *^1^ [231] PP2A activationBreast [232], gastric [233] and GBM [95]Inhibited pAkt,Increased apoptosistackled adriamycin and cisplatin resistancePre-clinicaltetracyclic triterpenes isolated from Cucurbitaceae and Cruciferae plantsErlotinib *^1^ (and other quinazoline derivates)PP2A activationHepatocellular carcinoma [234]Inhibits CIP2A-PP2A-pAkt axis resulting in decreased pAkt (independent of its classical role as a EGFR inhibitor)ClinicalSmall molecule kinase inhibitor*^1^: Compounds that can effectively cross the blood-brain barrier. *^2^: Compounds that cannot effectively cross the blood-brain barrier. Studies proving data on efficiency of crossing the blood-brain barrier is either cited or accessed from the RUGBANK drug database accessible via https://go.drugbank.com/drugs, accessed on 6 October 2022.


## 6. Concluding Remarks

PP2A has a very complex regulatory network given the multiplicity of holoenzyme composition and substrate specificity. The literature reflects a confounding duality in utilizing PP2A as a tool to fight cancer. Altogether, it seems that PP2A modulation in the context of cancer treatment needs to be considered cautiously as it is highly cancer type-, tissue type-, as well as tumor stage-dependent. The role of PP2A has been extensively analyzed in late stage tumors, but precancerous lesions have not been addressed. Individualized therapies using PP2A activity modulators seem to be attractive approaches for the increasing efficacy of chemotherapy and radiation therapy. In addition, the role of PP2A in therapy resistance needs to be further investigated. More systematic studies are also needed to better understand the immunomodulatory role of PP2A in immune-hot tumors as well as immune cell exclusion from cold tumors under hypoxia.

One of the caveats of using PP2A mimetics or PP2A inhibitors is that it is not possible to interfere specifically with holoenzymes containing one particular B subunit. Inhibition of only a restricted set of holoenzymes would allow to target pathways more specifically, potentially avoiding broad adverse side effects. For a better understanding of the distinct roles of PP2A holoenzymes, specific antibodies for individual subunits are needed [13,15]. Functional analyses by overexpression or knockdown of specific B subunits should be interpreted with caution, because the observed phenotype could be attributed to formation of secondary PP2A complexes formed by outcompeting other B subunit isoforms. Quantitative phosphoproteomics and interactomics approaches may prove to be useful for the identification of physiological targets of distinct PP2A complexes [235].

## Figures and Tables

**Figure 1 cancers-14-05227-f001:**
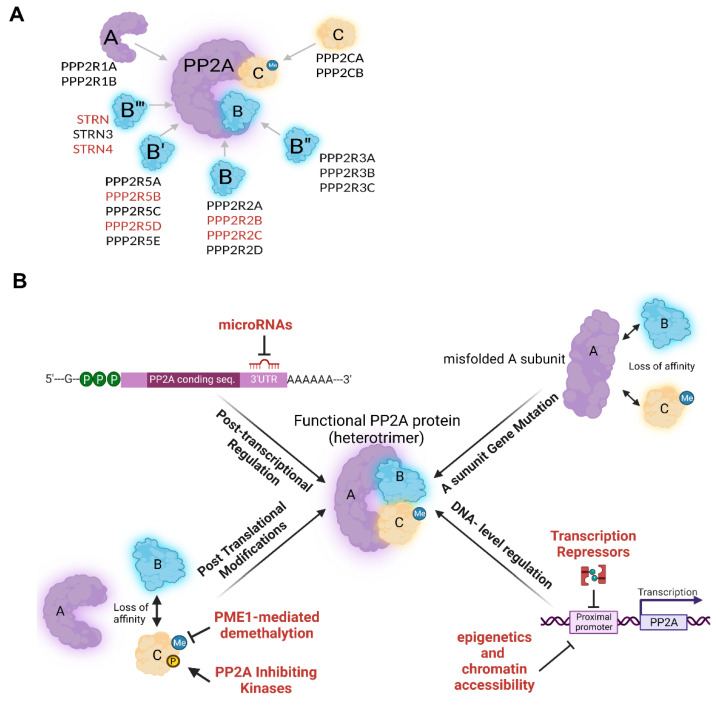
(**A**) PP2A heterotrimer composition consisting of structural (A), catalytic (C), and regulatory (B) subunits. The latter, has different subfamilies of B, B’, B’’ and B’’’ (each with 3–5 different isoforms). PP2A subunits which are more abundant in the brain tissue compared to other tissues are indicated in red. STRN = PPP2R6A; STRN4 = PPP2R6C. (**B**) Mechanisms of PP2A downregulation. Text in red indicates mechanisms that are more prevalent in cancer.

**Figure 2 cancers-14-05227-f002:**
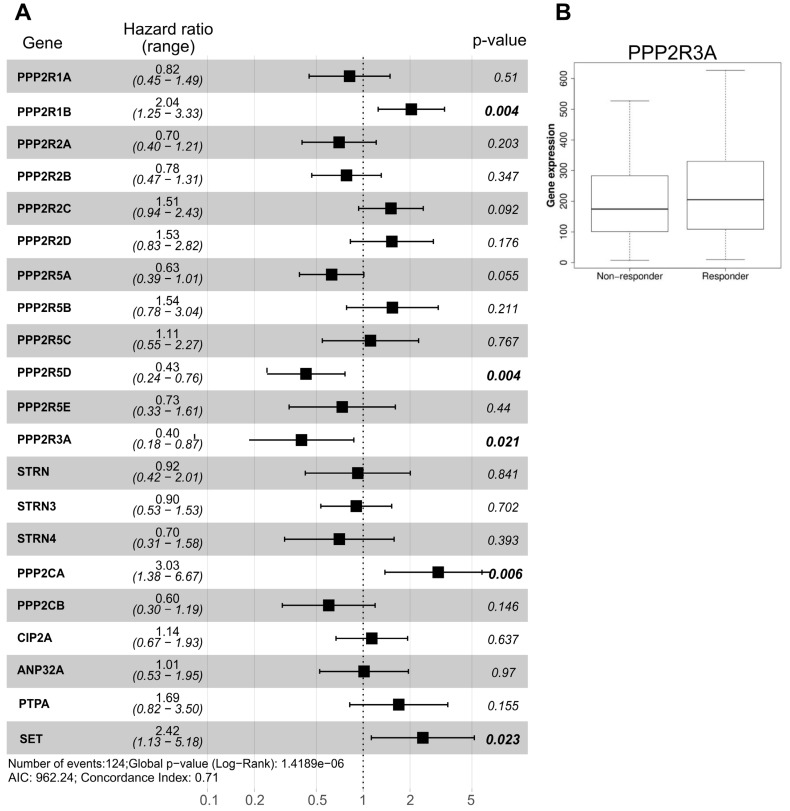
Association of PP2A subunits expression and GBM survival. (**A**) Hazard ratio based on the TCGA GBM dataset for each PP2A holoenzyme. For all the genes depicted, n = 156 TCGA-GBM cases were analyzed. Significant *p*-values are indicated in bold. (**B**) Analysis of ROC plotter indicating significant higher expression of PPP2R3A in TMZ responders (*p* = 0.042, n = 319: 154 non-responders and 165 responders).

**Figure 3 cancers-14-05227-f003:**
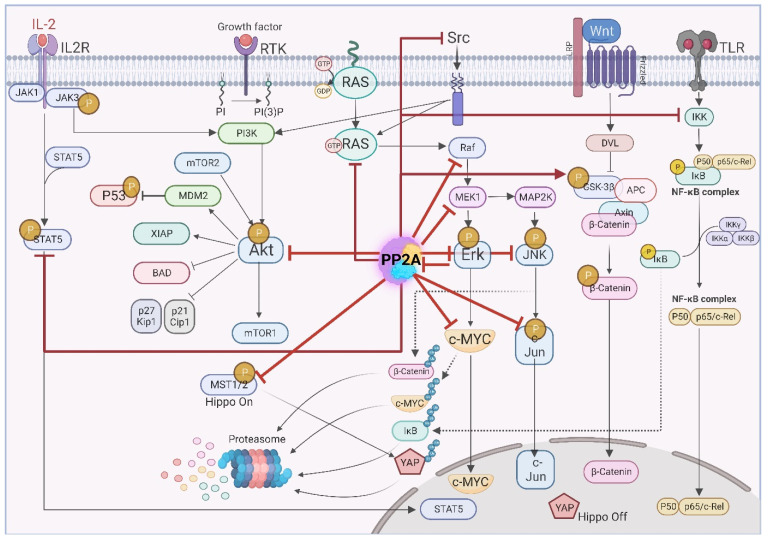
PP2A simultaneously regulates many cancer-associated pathways by targeting various aberrantly activated kinases.

**Figure 4 cancers-14-05227-f004:**
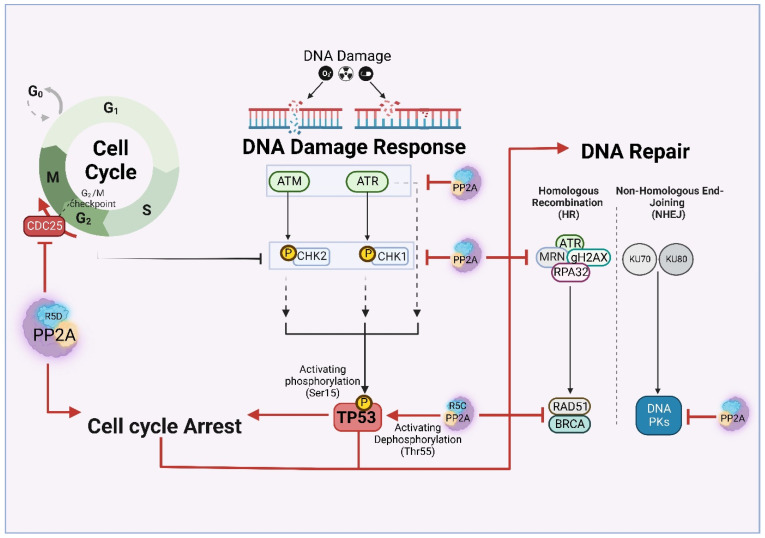
PP2A regulation interconnects cell cycle, DNA damage response and DNA repair pathways.

**Figure 5 cancers-14-05227-f005:**
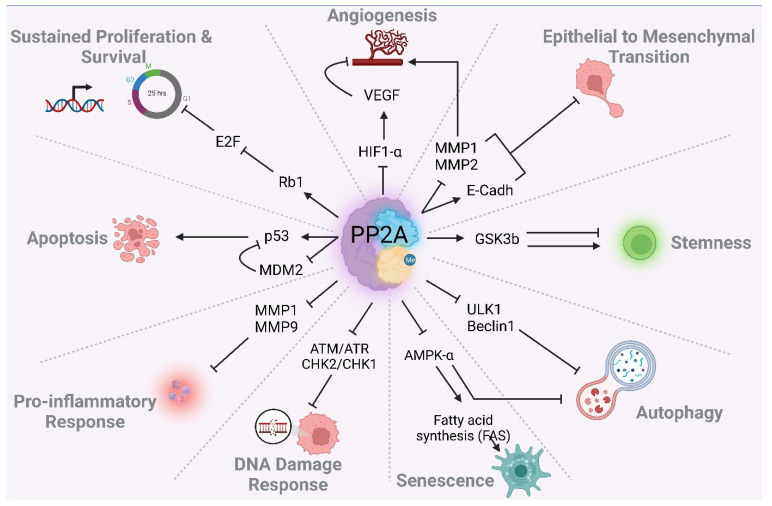
PP2A regulates cancer-associated cellular processes and cancer hallmarks.

**Table 1 cancers-14-05227-t001:** PP2A dysregulated subunits in various cancer types and their mechanism of deregulation.

Cancer Type	Dysregulated PP2A Subunit (and Abundance)	Mechanism of Deregulation	Functional Impact of the Alternation
Glioma	PPP2R1A (at least 50% of gliomas) [31]	Very rare mutations, Mostly post-translational	Supports tumorigenesis
	PPP2R1B [31]	No mutations reported, translational and post-translational only	Supports tumorigenesis
	PPP2R2C [35]	Decreased expression	Supports gliogenesis by increased mTOR expression
Lung cancer (NSCLC)	PPP2R1B (15%) [36,37]	Splice site mutation/Loss of heterozygosity (LOH)	Supports tumorigenesis
	PPP2R1A [38,39]	Mutation/LOH	Supports tumorigenesis
	PPP2R5E [27]	Post-transcriptional (by microRNA-19b)	Supports tumorigenesis and resistance to TKIs
	PPP2R5C [30]	Point mutation (loss of function)	Supports tumorigenesis by TP53 inactivation
Colorectal Cancer (CRC)	PPP2R1B (15%) [36,37,40]	Mutation/LOH	Supports tumorigenesis
	PPP2R5E	Decreased expression	Confers apoptosis resistance, supports tumorigenesis
	PPP2R2B [41,42]	DNA hypermethylation	Supports tumorigenesis by PDK1 & Myc deregulation + confers resistance to rapamycin [41]
	PPP2R1A [3]	Decreased expression	Supports tumorigenesis
Hepatocellular Carcinoma (HCC)	PPP2R1A [43]	NFκB-mediated promoter hypermethylation) [16]	Supports tumorigenesis
	PPP2R1B [43]	Alternative Splicing	Supports tumorigenesis
Breast cancer (BC)	PPP2R1B [38]	Splice site mutation/LOH	Supports tumorigenesis
	PPP2R1A [38,39]	Mutation/LOH	Supports tumorigenesis
	PPP2R2B (lowest expressed PP2A subunit in HER2+BC) [8]	EZH2-mediated hypermethalytion	Poor prognosis, poor HER2-targeted therapy response
	PPP2R2A [44]	Deleted in a subset of luminal BC (mitotic ER+)	Not clear
Melanoma	PPP2R1A (less frequent) [38,39]	Mutation	Not clear
	PPP2R1B [38]	Abnormal splicing	Supports tumorigenesis
	PPP2R5C [45,46]	Decreased expression	Supports melanoma development [45] & progression by dephosphorylating Paxillin at focal adhesions [46]
	PPP2R5A [47]	Decreased expression (lowest expression in metastatic melanoma [3])	Supports tumorigenesis by c-MYC protein stability and oncogene-induced senescence
Acute Myeloid Leukemia (AML)	PPP2R5E [48]	Decreased expression	Confers apoptosis resistance, supports tumorigenesis
	PPP2R2A [15]	Post-translational, rapid proteolysis [49]	Shorter complete remission, adverse prognosis (Introduced as specific pAkt regulator in AML [3])
	PPP2R1B [50]	Exon 9 deletion	Supports tumorigenesis
	PPP2CA [50]	Deletion (5q) (Mainly in p53-mutated tumors)	Supports tumorigenesis
Ovarian cancer	PPP2R1A [51,52,53]	Somatic mutation	Supports tumorigenesis
	PPP2R1B [3]	Loss of heterozygosity	Supports tumorigenesis
Prostate cancer (PCa)	PPP2R2A [54]	Homozygous deletion	Associated with incidence of PCa
	PPP2R2C [7,55]	Decreased expression	Associated with metastasis, PCa Specific Mortality (PCSM) and castration-resistance
	PPP2CA [56]	Decreased expression (especially in androgen insensitive prostate cancer [57])	Correlates with tumor stage and Gleason grade
Sarcoma	PPP2R5E [58]	SNP (mutation)	Associated with soft tissue sarcoma onset & worse overall survival
Myeloma	PPP2R2A [59]	Deletion	Supports tumorigenesis

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
