# Peer review of "Pleiotropy of PP2A Phosphatases in Cancer with a Focus on Glioblastoma *IDH* Wildtype"

_cancers, 2022, doi:10.3390/cancers14215227_

Round 1
Reviewer 1 Report
In their manuscript, Kashani and Vasella review the mechanistic role and potential therapeutic vulnerability of Serine/Threonine protein phosphatase 2A (PP2A) in cancer, in general, and IDH-wildtype glioblastoma (GBM), in particular. The authors have gone to great lengths to discuss a large number of pertinent topics associated with this enzyme in cancer therapy. They begin with chapters on PP2A composition, regulation, and dysregulation. A stand-alone section on PP2A dysregulation in glioma highlights current findings in the GBM realm and is followed by a more detailed review of PP2A-regulating molecules. After this, the authors summarize oncogenic signaling pathways impacted by PP2A. This is followed by a section on the distinct role of PP2A in various cellular processes such as cell cycle progression, DNA repair, apoptosis, EMT, senescence, etc. Finally, as a subsection of “Role of PP2A in cellular processes”, therapeutic approaches targeting PP2A are reviewed, including a paragraph on the potential use of PP2A restoration for overcoming TMZ resistance in GBM.
Overall, this review is well written and designed. It covers a clear medical need in oncology. While reviews of PP2A in cancer therapy are not necessarily novel (Perrotti et al., 2013; Ruvolo et al., 2016; to name a few), this research is novel in its intended effort to focus on GBM. There are a few concerns/issues in the manuscript the authors should address:
Major concerns:
1. The overall structure of the manuscript is a little confusing, it does not follow a logical flow. I highly recommend re-ordering certain sections and numbering all sections and sub-sections to facilitate an easier understanding for the reader. Some sections are redundant (as indicated by the authors, “see above” etc.) and may be deleted by restructuring the article. I specifically want to highlight that it will be important to make the review of preclinical and clinical therapeutic approaches its own section, with a separate sub-section focusing on PP2A therapies for GBM (the discussion about TMZ resistance and PP2A may be a part of this).
2. Since the manuscript is being submitted to a special issue on “New Approaches with Precision Medicine in Brain Tumors”, it is required to put strong emphasis on this aspect of PP2A. While a general review of basic enzyme properties and discussion across cancer types is well-received, the focus should be on GBM – throughout the whole manuscript (including the abstract).
Minor concerns:
1. The following sentences lack references (line #): 49, 124, 149, 290, 294 (whole paragraph), 338, 352, 353, 359, 364, 529, 537, 548, 551, 594, 629 (reference clinical trials).
2. The statement in line 102-103 should be moved in front of the corresponding section in the text and highlighted in Figure 1.
3. Line 148-149: Refer to the 2021 WHO classification of Tumors of the Central Nervous System (Louis et al., 2021). Presumably, the authors mean glioblastoma, IDH wild-type here. Even though this has been defined in the abstract, it should be defined again upon its first appearance in the manuscript body.
4. Line 155: “GBM, IDH wild-type” is redundant if previously defined as GBM.
5. Line 184: Should read figure 2A.
6. Line 192: Should read figure 2B.
7. Line 147-194 and figure 2: As the regulation of PP2A on the post-transcriptional level seems to be of importance, it would be worthwhile to include published data on protein levels here, if available.
8. Line 200: Implication in GBM?
9. Line 261: Rename as inhibitory proteins, otherwise misleading.
10. Line 609: Discuss patient data here (including GBM), references.
11. Line 631: This may be favorable for systemic delivery of therapeutics across the BBB to brain tumors, reference reports...
12. Line 632: Which trials?
13. Line 645: Remove duplicate (“base excision repair (NER)”).
Reviewer 2 Report
In their manuscript entitled ‘Pleiotropy of PP2A phosphatases in cancer with a focus on glioblastoma IDH wildtype’, the authors review the biological heterogeneity and function of PP2A phosphatases. The review is comprehensive, encompassing the subunit/isoform breadth, regulation of PP2As, and functional consequences of PP2A activation/deactivation. The manuscript is well-written and well-cited.
Major Concerns:
1. I was initially very excited by the manuscript title and significance section, anticipating a review on PP2As in glioblastoma (GBM). The concept is very novel! However, the bulk of the manuscript is general information on PP2As, with only a few scattered sections on PP2A biology in GBM. What is known about PP2As modulating oncogenic signaling pathways and/or cellular processes in GBM? This information seems critical to a review focusing on PP2As in GBM. If this information is not known, I would recommend changing the title to suggest a more general review of PP2A in cancer.
2. The 3rd paragraph of the first section discusses the heterogeneity in PP2A subunits and isoforms. This section is dense and difficult to distill down into take-away points. Can the author’s present some of this information visually in a graphs or chart? What is the important message here? How does this relate to GBM?
Minor Concerns:
1. Figure 2: The unbolded font is difficult to read.
2. Acronyms are not initially defined: DDR used initially line 55, PP2A used initially line 63.
3. Line 84: missing a word after ‘compete’
4. Line 316: missing a space after ‘in’
5. This kind of analysis is very novel and I would guess that many readers are not used to reading these kinds of figures. Demonstrating figures of global efficiency and cost across all three patient populations is recommended for clarity.
Reviewer 3 Report
This is a well written and comprehensive overview on the role of PP2A in cancer, focusing on IDH-wildtype glioblastoma. The authors convincingly show that these phosphatases are not just unspecific enzymes that simply turn off signaling cascades activated by kinases, but carry several distinct roles in oncogenesis. Especially their ability to modulate the efficacy of chemotherapy and radiotherapy I find intriguing. I have a few general and minor comments with suggestions for further approvement of the paper:
General comments:
1. In the introduction, a background is lacking on the classification of phosphatases, i.e., how they are grouped by substrate specificity into different serine/threonine, tyrosine phosphatases. It would also be worthwhile to mention the classification of Ser/Thr phosphatases into subfamilies of PP1, PP2A, PP2B, PP4, PP5, PP6, PP7 etc., which is not clearly stated. Please explain why the focus of this review is on the PP2A family member?
2. I would like to suggest removing Figure 2, that does not add any value to the paper. It is already obvious from the text that the different subunits are expressed at various levels and regulated by complex mechanisms. Providing simple Hazard ratios for survival (without testing these in a model together with established prognostic factors such as patient age, performance status, molecular tumor profile, type of treatment etc.) is not meaningful.
3. With respect to GBM treatment, it would be of interest to add a new column to Table 2, showing which compounds are able to cross the blood-brain barrier. (Fingolomid, for example, crosses the BBB and is used in the treatment of relapsing-remitting multiple sclerosis).
Minor comments:
There are some misleading typos in the manuscript, for example:
1. In the Abstract, first word: seine instead of serine
2. Line 150: PP2A-decativating instead of PP2A-deactivating
